# Deep microbial proliferation at the basalt interface in 33.5–104 million-year-old oceanic crust

Yohey Suzuki[1,7 ✉], Seiya Yamashita[1,7], Mariko Kouduka[1,7], Yutaro Ao[1], Hiroki Mukai[1,6], Satoshi Mitsunobu [2], Hiroyuki Kagi[3], Steven D'Hondt [4], Fumio Inagaki [5,6], Yuki Morono [5], Tatsuhiko Hoshino[5], Naotaka Tomioka [5] & Motoo Ito [5]

The upper oceanic crust is mainly composed of basaltic lava that constitutes one of the largest habitable zones on Earth. However, the nature of deep microbial life in oceanic crust remains poorly understood, especially where old cold basaltic rock interacts with seawater beneath sediment. Here we show that microbial cells are densely concentrated in Fe-rich smectite on fracture surfaces and veins in 33.5- and 104-million-year-old (Ma) subseafloor basaltic rock. The Fe-rich smectite is locally enriched in organic carbon. Nanoscale solid characterizations reveal the organic carbon to be microbial cells within the Fe-rich smectite, with cell densities locally exceeding $10^{10}$ cells/cm$^3$. Dominance of heterotrophic bacteria indicated by analyses of DNA sequences and lipids supports the importance of organic matter as carbon and energy sources in subseafloor basalt. Given the prominence of basaltic lava on Earth and Mars, microbial life could be habitable where subsurface basaltic rocks interact with liquid water.

[1] Department of Earth and Planetary Science, The University of Tokyo, 7-3-1 Hongo, Bunkyo-ku, Tokyo 113-0033, Japan. [2] Department of Environmental Conservation, Graduate School of Agriculture, Ehime University, 3-5-7 Tarumi, Matsuyama, Ehime 790-8566, Japan. [3] Geochemical Research Center, The University of Tokyo, 7-3-1 Hongo, Bunkyo, Tokyo 113-0033, Japan. [4] Graduate School of Oceanography, University of Rhode Island, 215 South Ferry Road, Narragansett, RI 02882, USA. [5] Kochi Institute for Core Sample Research, Japan Agency for Marine-Earth Science and Technology (JAMSTEC), Monobe B200, Nankoku, Kochi 783-8502, Japan. [6] Present address: Mantle Drilling Promotion Office, Institue for Marine-Earth Exploration and Engineering, JAMSTEC, Showa-machi 3173-25, Kanazawa-ku, Yokohama 236-0001, Japan. [7] These authors contributed equally: Yohey Suzuki, Seiya Yamashita, Mariko Kouduka. ✉email: yohey-suzuki@eps.s.u-tokyo.ac.jp

The upper oceanic crust is mainly composed of basaltic lava[1,2]. It has been continuously created on Earth for ~3.8 billion years[3]. Basaltic lava is erupted and solidified at mid-ocean ridges where high-temperature basalt-seawater reactions provide substantial energy for sustaining chemosynthetic life[4]. On ridge flanks, circulation of crustal fluid is hydrothermally driven within the basaltic lava overburdened with sediments[2]. The portion of basaltic lava beneath sediment cover is referred to as basaltic basement. Previous studies at 3.5- and 8-million-year-old (Ma) ridge-flank systems demonstrated that these young crustal aquifers, respectively, harbor anaerobic thermophiles and aerobic mesophiles that contribute to hydrogen, carbon, and sulfur cycling[5–7]. After rock fractures are filled with secondary minerals, intensities of fluid circulation and basalt-seawater reactions sharply decline with increasing crustal age; with most crustal oxidation occurring in the first 10 million years after crust formation[8].

More than ca. 90% of Earth's ocean lithosphere is older than 10 Ma[9] and long past its early stage of relatively high crustal oxidation rate. Despite its vast areal extent, the nature and extent of life in this old crust is previously unknown, in part because of the technological and analytical challenges of exploring the igneous rock habitat through scientific drilling[10]. Alteration textures suggestive of biological activity have been observed in oceanic crust as old as 3500 Ma[11]. However, the role of microbial activities in creating these textures and the age of the crust at the time of texture formation remain unknown.

Here, we investigate the occurrence of microbial communities in subseafloor basaltic lava older than 10 Ma, recovered by Integrated Ocean Drilling Program (IODP) Expedition 329 in the South Pacific Gyre (SPG). The presence of microbial cells in the iron-rich smectite on old subseafloor basaltic rock was revealed by nanoscale solid characterizations. Analysis of their lipid profiles and DNA sequences reveals the dominance of heterotrophic bacteria, suggesting the presence of organic matter resources in the subseafloor basalt.

## Results

**Smectite-hosted microbial life**. Within the SPG, extremely low sedimentation rates lead to burial of sediments nearly depleted in organic matter[12] (Supplementary Table 1). In this ultra-oligotrophic environment, dissolved $O_2$ penetrates from the ocean floor to the basaltic basement and sustains aerobic microbes throughout the sediment column[7] (Supplementary Fig. 1). During Expedition 329, using the drilling vessel *JOIDES Resolution*, core samples were obtained from basaltic basement at Sites U1365, U1367, and U1368 with crustal ages of 104 Ma[13], 33.5 Ma[14], and 13.5 Ma[14] (Supplementary Table 1).

Mineral characterizations were conducted for core samples with fractures/veins to clarify the presence of clay minerals typically produced by low-temperature rock–water interactions (weathering). X-ray diffraction analysis revealed the presence of Fe-rich smectite in 33.5-Ma and 104-Ma core samples but not in 13.5-Ma core samples[15]. Thin sections were prepared from the 33.5-Ma and 104-Ma core samples with sample codes: U1367F-6R1, U1365E-8R4, and U1365E-12R2 at depths of 51, 109.6, and 121.8 m below the seafloor (mbsf) and observed by scanning and transmission electron microscopies (SEM and TEM) coupled to energy-dispersive X-ray spectroscopic (EDS) analysis. Fe-rich smectite was found at the rims of fractures and veins mainly filled with celadonite and iron oxyhydroxides in U1365E-8R4 and U1365E-12R2, respectively, whereas veins are filled with Fe-rich smectite in U1367F-6R1[15]. Two types of compositionally distinct Fe-rich smectite veins were observed in U1367F-6R1: one is similar to those found in U1365E-8R4 and U1365E-12R2 with high Mg and K contents; the other is characterized by high Fe content, as typically observed in Fe-rich

smectite from deep-sea hydrothermal mounds (Supplementary Fig. 2a and Supplementary Table 2).

Fluorescence microscopy observations of the thin sections reveal that SYBR Green I-stained cell-like fluorescence signals are extensive along the rims of the rock fractures/veins associated with Fe-rich smectite in U1365E-8R4 and U1365E-12R2 (Fig. 1). Although Fe-rich smectite with high Mg and K contents in U1367F-6R1 is correlated with fluorescence signals (Supplementary Fig. 2b), fluorescence signals were not detected from veins filled with Fe-rich smectite with high Fe content in U1367F-6R1 (Supplementary Fig. 2c).

To confirm that these greenish signals originate from microbial cells rather than from autofluorescent materials, ~10 × ~10-μm² sections with a thickness of ~3 μm were fabricated by focused ion beam (FIB), and element-mapping images were obtained using nanoscale secondary ion mass spectrometry (NanoSIMS). FIB-NanoSIMS analysis of U1365E-8R4 revealed overlapping signals of $^{12}C^{14}N^-$, $^{31}P^-$, and $^{32}S^-$ on the dense spots stained with SYBR Green I, indicating that those greenish signals are derived from microbial cells (Fig. 2). The microbial cells are localized in the proximity of microscale voids and enrobed within Fe-rich smectite[16].

The same result was obtained by FIB-NanoSIMS of U1365E-12R2 (Supplementary Fig. 3). Element mapping using scanning transmission electron microscopy (STEM) equipped with energy dispersive spectroscopy (EDS) showed that the microbial cells are spatially associated with laths of Fe-rich smectite (Supplementary Fig. 3). Given this association and the large compositional difference between Fe-rich smectite and the bentonite clay used for drilling mud, the microbial cells were not introduced from the drilling mud (Supplementary Fig. 4 and Supplementary Table 2). These results indicate that the detected signatures along the mineral-filled fractures/veins are derived from indigenous microbial communities in the deep crustal biosphere beneath the oceanic and sedimentary biospheres.

**Microbial community composition**. Core samples were evaluated for contamination using fluorescence microspheres (0.5 μm in diameter) that mimic microbial cells introduced from drilling fluid[17] (Supplementary Fig. 5). Microscopic counting of microspheres in subsamples before and after cleaning steps such as washing with 3% NaCl solution and flaming the exterior showed that untreated exteriors of core samples contained detectable microspheres, but most post-treatment sample interiors contained no detectable microspheres (Supplementary Fig. 6 and Supplementary Table 3). These results clarify that the contamination evaluation was properly conducted to show the level of drilling contamination for DNA analysis. 16S ribosomal RNA gene sequences were obtained from the V4 to V6 regions by tag-sequencing from four core samples with no detected microspheres (U1365E-8R4, and -12R2 and U1367F-4R1 and U1368F-4R2), one microsphere-detected sample (U1368F-7R3), drilling fluid used at Site U1365, and a DNA extraction blank. To identify potentially contaminant OTUs from drilling and subsequent laboratory manipulations, the highly contaminated core from U1368F-7R3, the drilling fluid from U1365E, and the negative control were compared to the microsphere-undetected samples.

Because α- and β-proteobacterial OTUs were identical among the contaminated sources such as the drilling fluid and the DNA extraction blank, the OTUs detected from the contaminated sources were removed from the microsphere-undetected core samples (Supplementary Fig. 7 and Supplementary Data 1, 2). In addition, OTUs obtained from the highly contaminated core (U1368F-7R3) were excluded for detailed analyses of indigenous microbial communities. According to phylogenetic affiliation

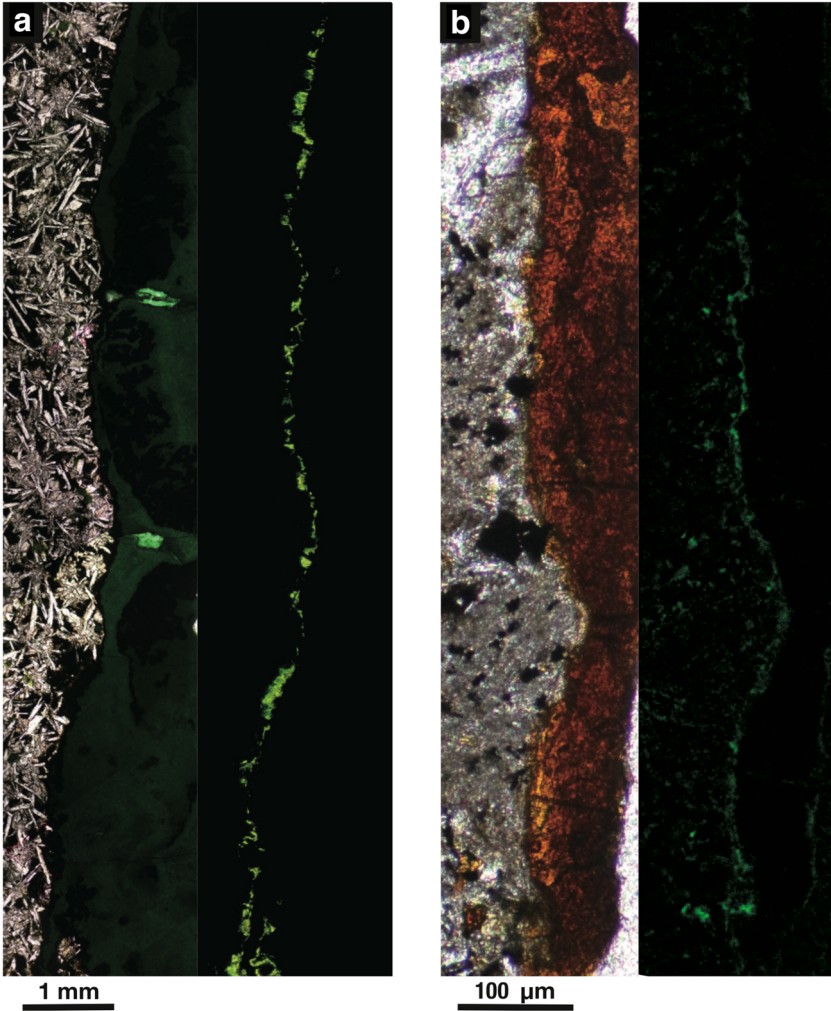

**Fig. 1 Basalt interface with microbial colonization.** Light and fluorescence microscopy images of SYBR Green I-stained microbial cells in a fracture filled with celadonite in of U1365E-8R4 (**a**) and in a vein filled with iron oxyhydroxides in U1365E-12R2 (**b**).

based on 16S rRNA gene sequences (Fig. 3), three types of microbial communities were identified, as outlined below.

- Type SPG-I (relatively young crustal community: 13.5 Ma). At Site U1368, γ- and ε-proteobacterial sequences were proportionally abundant and included strains related to the genera *Arcobacter, Thioreductor, Sulfurimonas,* and *Sulfurovum* known as deep-sea sulfur- and/or hydrogen-oxidizing chemolithoautotrophs[16] and the genus *Alteromonas* globally distributed in deep-sea aquatic habitats with aerobic heterotrophy[18] (Fig. 3 and Supplementary Data 2).
- Type SPG-II (aged crustal communities: 33.5–104 Ma). At Sites U1365 and U1367, β-proteobacterial sequences were predominant and closely related to aerobic organotrophs, such as *Roseateles depolymerans* isolated from pumice-bearing lake sediment[19] (Fig. 3 and Supplementary Data 2).
- Type SPG-III was only observed in U1365E-12R2 (a depth of 122 mbsf), in which γ-proteobacterial sequences affiliated within the family Methylococcaceae were predominantly detected (Fig. 3 and Supplementary Data 2). In general, these crustal microbial communities were comprised of Methylococcaceae members typically found in methane-rich fluids emitted from the deep ocean floor, including cold seeps and hydrothermal vents[20].

Microbial communities have previously been observed in rock core and fluid samples from North Pond (North Atlantic IODP Site; 8 Ma), where oxygenated cold fluid actively circulates in sediment-covered basaltic basement[7]. For microbiological investigations, a Circulation Obviation Retrofit Kit (CORK) was installed to collect fluid samples from the basaltic basement at North Pond. However, it is possible that microbial communities in fluid samples are distinct from those attached to adjacent rock surfaces. Microbial communities in the North Pond fluid samples are mainly comprised of members of *Campylobacterales* and *Alteromonadales* (Fig. 3)[21]. In contrast, rock core samples were not dominantly colonized by *Campylobacterales* members but *Alteromonadales* members[22]. Dominant microbial populations obtained in our rock sample from Site U1368 (13.5 Ma; Fig. 3) were similar to those obtained from the North Pond fluid samples. Thus, the nature of the basement fluid may be very similar in relatively young (8 Ma and 13.5 Ma) basaltic basement in both the Atlantic and Pacific Oceans. These results also indicate that the crustal biosphere can be technically evaluated from rock cores, as well as from circulating fluid.

**Environmental controls on community composition.** Basement at 13.5 Ma and 33.5 Ma is mainly composed of pillow lava covered

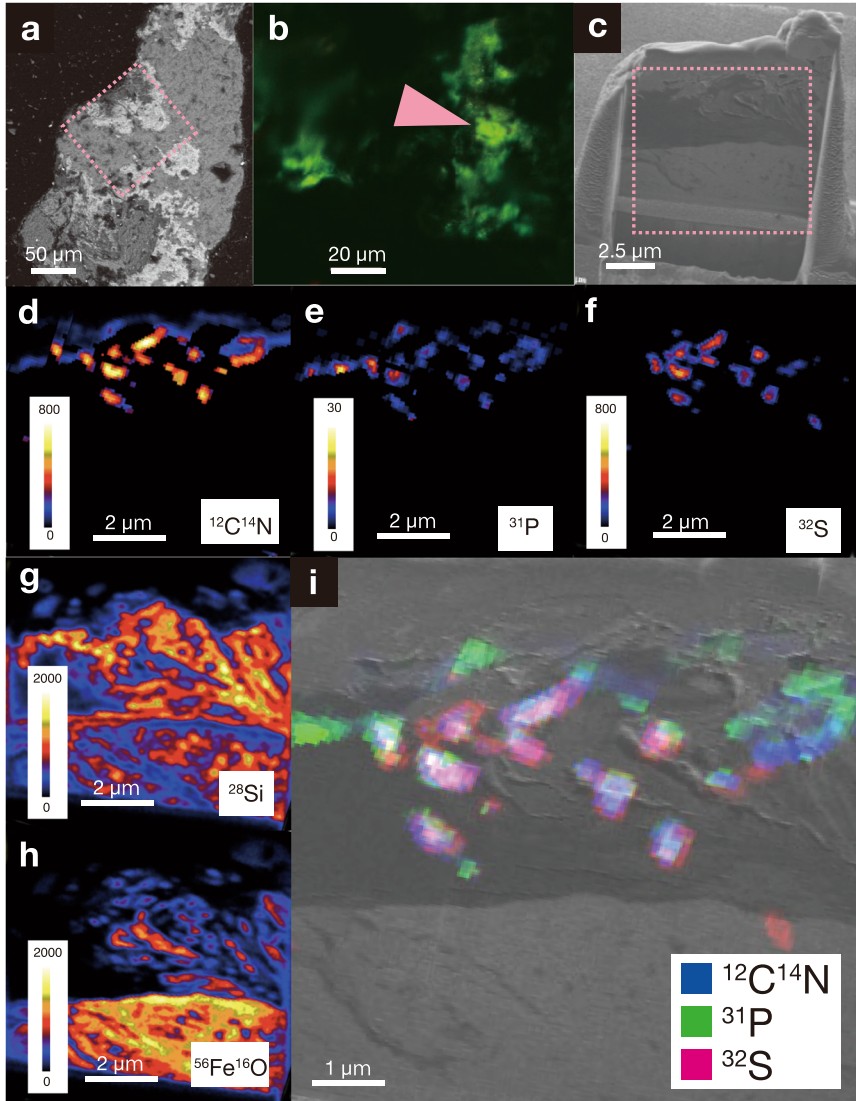

**Fig. 2 Single-cell characterizations of fracture-hosted microbial populations.** Scanning electron microscopic (SEM) image of a mineral-filled fracture in U1365E-8R4 (**a**). Confocal laser microscopy image of SYBR Green I-stained microbial cells (**b**). SEM image of a FIB-derived thin section of U1365-8R4 with a square region (~10 × ~10 μm$^2$) analyzed by the JAMSTEC NanoSIMS (**c**). NanoSIMS images of $^{12}C^{14}N^{-}$ (**d**), $^{31}P^{-}$ (**e**), $^{32}S^{-}$ (**f**), $^{28}Si^{-}$ (**g**), and $^{56}Fe^{16}O^{-}$ (**h**) with intensity color contours. Overlays are shown from the Ga ion image of the FIB section in black and white and the NanoSIMS images of $^{12}C^{14}N^{-}$ in blue, $^{31}P^{-}$ in green, and $^{32}S^{-}$ in red (**i**). Dashed rectangles and an arrow show regions presented in the following figures.

with 12- to 17-m thick sediment (Supplementary Fig. 1 and Supplementary Table 1). The deepest sediment at both sites contains similar concentrations of dissolved $O_2$ and dissolved nitrate. Although the crustal structure and the dissolved oxidant chemistry are fairly similar at both sites, microbial community composition differs notably between the 13.5- and 33.5-Ma basements (Fig. 3). The clay minerals that form in fractures/veins by low-temperature rock-water interactions provide information that may explain the difference between these communities[15]; the presence and absence of Fe-rich smectite in fractures/veins at Sites U1367 and U1368 indicate that Fe-rich smectite formation was inhibited by vigorous seawater circulation at U1368[16] (Fig. 4).

Although the basaltic basement at Site U1365 comprises lava flows where fluid flow is generally between sheeted layers rather than along chilled margins of pillow lava[23], its microbial community composition is similar to that found at Site U1367, which is consistent with the presence of Fe-rich smectite at Sites U1365 and U1367. Seafloor heat flow at U1367 and U1365 is

consistent with conduction as the dominant mode of heat transport, while heat flow at U1368 falls below the expected conduction-only level, consistent with apparent heat transport by fluid circulation within the rocky crust[23]. This difference is consistent with the basement ages of the respective sites, as fluid circulation and advective heat transport are generally much more vigorous in relatively young, warm crust (such as the 13.5-Ma crust at U1368) than in much older and consequently cooler crust (such as the 33.5-Ma and 104-Ma crust at, respectively, U1367 and U1365). We suggest that the basement habitability is controlled by heat and fluid flows, which generally decrease over time, versus the primary structure of the crust (e.g., pillow basalt or flow basalt). In addition, the formation of Fe-rich smectite in the basaltic basement appears to be correlated with the kinds of microorganisms in aged oceanic crust (Fig. 4).

**Crustal biosphere fuelled by mineral-bound organic matter.**
Observations of microbial cells in FIB sections (10 μm × 10 μm ×

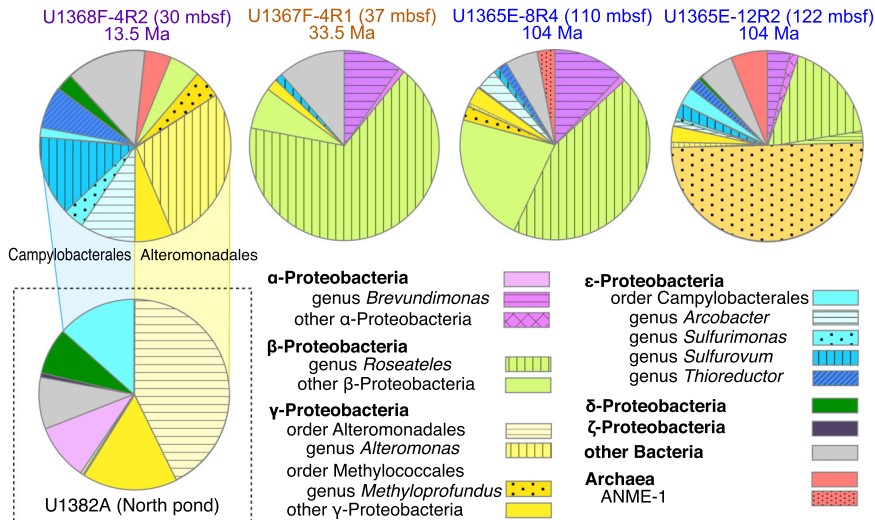

**Fig. 3 Community composition in cold basaltic basement based on 16S ribosomal RNA gene sequences.** Taxonomic profiles of basaltic rock cores from the South Pacific Gyre are shown as pie charts with major taxonomic groups ranging from genus to phylum. Community composition of a crustal fluid sample at North Pond (Site U1382; 8 Ma) is shown as a pie chart for comparison[21].

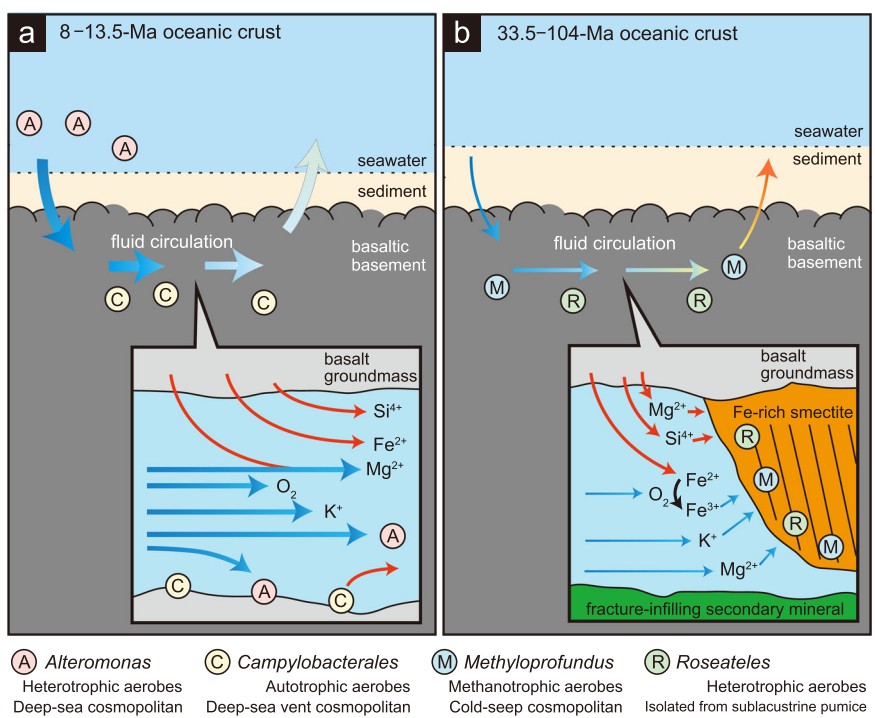

**Fig. 4 Schematic illustrations of fluid flow regimes and key microbial populations.** In basaltic basement, substrates are distinctively supplied from seawater and basalt rocks in 8–13.5 Ma (**a**) and 33.5–104 Ma (**b**). Blue and red arrows indicate abundant substrate supplies from seawater and basalt rocks, respectively.

3 μm) suggest a cell density range of $n \times 3.3 \times 10^9$ cells/cm$^3$, where $n$ represents a number of cells detected in a FIB section. In the FIB sections, 15 and 2 CN-bearing spots derived from microbial cells are visualized in U1365E-8R4 and U1365E-12R2, which gives approximate cell numbers of 5.0 and $0.7 \times 10^{10}$ cells/cm$^3$. This cell density is narrowly limited to the Fe-rich smectite at the interface between basalt and alteration minerals. Within that interface, cell density is exceedingly high in comparison with cell density in the deepest sediment overlying the basaltic basements at Sites U1365 and U1367[14] (~$10^2$ cells/cm$^3$), and in comparison with low-temperature fluids collected from 8-Ma basalt basement at North Pond[23] (~$10^4$ cells/cm$^3$). The

range of cell density estimated for the Fe-rich smectite of the basalt-water interface is nearly the same or higher as in organic-rich near-seafloor sediment deposited on continental margins[24].

To verify the cell density estimates in the two FIB sections, μ-Raman spectroscopy was used to obtain a diagnostic spectrum from the microbe-smectite assemblage (Fig. 5). The spectrum is composed of broad peaks at 1200–1600 cm$^{-1}$ attributed to amorphous organic matter and a slope increasing with Raman shift attributed to smectite[15]. The fingerprint spectrum was obtained throughout the interface regions filled with Fe-rich smectite with high Mg and K contents in U1367F-6R1, U1365E-8R4, and U1365E-12R2, but not from that filled with Fe-rich

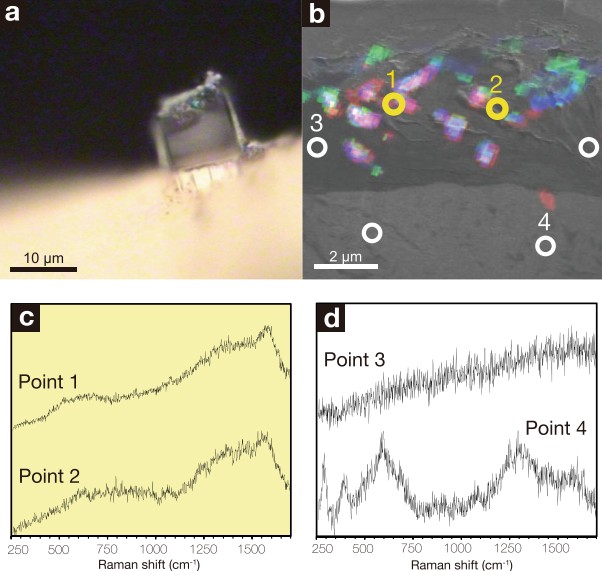

**Fig. 5 μ-Raman spectra of the assemblage composed of microbes and Fe-rich smectite in U1365E-8R4.** Optical microscopic image of the FIB section with a laser spot where a μ-Raman spectrum was obtained (**a**). Circles indicate points analyzed by μ-Raman spectroscopy in the Ga ion image overlain with NanoSIMS images (**b**). μ-Raman spectra from yellow circles associated with microbial cells (**c**) and from white circles without microbial cells (**d**).

smectite with the high Fe content in U1367F-6R1 (Supplementary Fig. 8). The lack of the fingerprint spectrum from Fe-rich smectite with high Fe content may be due to its formation at a deep-sea hydrothermal mound near the mid-ocean ridge.

Smectite is a fine-grained clay mineral, with a large surface area to adsorb organic matter[25]. As dominant microbial communities detected from 33- and 104-Ma basaltic basements are heterotrophic, it is conceivable that organic matter bound to Fe-rich smectite may help to sustain the high cell density at the basalt interface. Clay fractions were separated from the core samples and their organic carbon content was quantified. The clay fractions mainly composed of Fe-rich smectite contained up to 22-fold higher organic carbon than the bulk core samples (Supplementary Table 4), supporting the inference that mineral-bound organic matter fuels heterotrophic activities of microorganisms at the basalt interface. Fourier transform infrared-ray (FT-IR) spectra were obtained from the clay fractions to clarify the presence of lipids, based on the aliphatic $CH_3/CH_2$ absorbance ratios ($R_{3/2}$). Given that the $R_{3/2}$ values are domain-specific: Eukarya 0.3–0.5, Bacteria 0.6–0.7 and Archaea 0.8–1.0[26,27], the $R_{3/2}$ ranges of the clay fractions from U1365E-8R4, U1365-12R2, and U1367F-6R1 were approximately in the bacterial range (Supplementary Fig. 9), which agrees with the dominance of bacteria indicated by 16S rRNA gene sequences from the corresponding core samples U1365E-8R4 and U1365-12R2 and from the other core sample collected from the same site (U1367F-4R1).

16S rRNA gene sequences related to aerobic and anaerobic methanotrophs were prominent among the 16S rRNA gene sequences detected from the mineral-filled fractures in 104-Ma basaltic basement (Fig. 3). Almost half of the 16S rRNA gene sequences analyzed from U1365E-12R2 were closely related to *Methyloprofundus sedimenti*, an aerobic methanotrophic bacterium isolated from a deep-sea sediment sample associated with a whale fall[22]. Additionally, anaerobic methane-oxidizing archaea subtype 1 (ANME-1) was detected from U1365E-8R4 (Fig. 3). As

methane concentrations are below the detection limit (<1.3 μM) in all sediment samples at Site U1365[23], methane bound to Fe-rich smectite might be a source of energy for their persistence in situ[28].

## Discussion

The results of this study greatly extend understanding of bioenergetics and habitability in Earth's upper oceanic crust. Previous studies of bioenergetics in subseafloor basalt have generally focused on chemoautotrophic mineral oxidation, which mostly occurs in crust younger than ca. 10 Ma[29]. Our results indicate that cells encased in Fe-rich smectite densely coat rock surfaces of much older (33.5 Ma and 104 Ma) basalt and are largely sustained by aerobic heterotrophy and methanotrophy. Organic matter that may sustain these communities in the upper crustal aquifer includes (i) dissolved organic matter (DOM) in the seawater that flows through the fractures and veins[30], and (ii) organic matter abiotically synthesized during rock weathering (e.g, Lost City where amino-acid production associated with formation of Fe, Mg-rich smectite in gabbroic basement at the Lost City hydrothermal field[31]).

These results also have important implications for understanding the abundance and global distribution of microbial cells in the upper oceanic crust. Mineral (iron and sulfur) oxidation rates are highest in crust younger than ca. 10 Ma[32]. The number of cells that might be supported by aerobic iron oxidation in the upper marine crust has been estimated as $2.4 \times 10^{28}$ cells[29], potentially equivalent to 10% of total cell abundance in marine sediment[29] Because the abundant microbes reliant on aerobic heterotrophy and methanotrophy reside in much older crust (33.5 Ma and 104 Ma), inclusion of these heterotrophic and methanotrophic cells may substantialy increase estimate of total cell abundance in the upper oceanic crust.

The results of this study also have implications for the possibility of life on Mars and other planetary bodies. Basaltic crust is ubiquitous on other planets, such as Mars[33], as well on Earth. The Martian basaltic crust formed 4 billion years ago, to be followed by formation of Fe, Mg-rich smectite via hydrothermal alteration and weathering at the surface and in the subsurface until ~3 billion years ago[34,35]. On modern Mars, the surface is cold and dry under high vacuum conditions, and methane is emitted from the subsurface into the atmosphere[36]. Recently, the presence of subsurface liquid water has been indicated[37], which spurred international interest in the search for extraterrestrial life[38]. Given the subsurface presence of methane and liquid water on Mars, the communities fueled by organic matter and methane in subseafloor basalt on Earth provide a clear model for extant life and/or biosignatures from past life in the subsurface of Mars and other planets.

## Methods

**Sampling sites.** Core samples were collected from basaltic basement at Sites U1365, U1367, and U1368 in the South Pacific Gyre during Integrated Ocean Drilling Program (IODP) Expedition 329 (October 9 through December 13, 2010; Supplementary Fig. 1 and Supplementary Table 1). Although surface heat-flow data from Sites U1365 and U1367 were consistent with those expected for conductive crust in the absence of advection, data from Site U1368 showed substantially low heat flow, indicating apparent heat loss and advective circulation of overlying seawater within the oceanic crust (Expedition 329 Scientists, 2011d). The thickness of sediment cover ranged from 6 m to 71 m (Supplementary Table 1). Molecular oxygen ($O_2$) and nitrate penetrate through the sediment column from the overlying seawater to the basaltic basement, because sediment accumulation rates are so low that detrital organic matter is largely oxidized at the seafloor and $O_2$ diffusion rate outpaces organic oxidation rate in more deeply buried sediment (Supplementary Fig. 1 and Supplementary Table 1).

**Drilling and petrological description of basaltic cores.** Basalt samples were obtained using a rotary core barrel (RCB) coring system from boreholes U1365E,

U1367F, and U1368F. Although contamination of RCB cores from seawater-based drilling fluids is unavoidable, its extent can be carefully monitored. After each core recovery, all core sections were immediately transferred from the catwalk deck to the cold room on the hold deck of the drilling research vessel JOIDES Resolution. Prior to microbiological sampling, preliminary description of petrological characteristics was visually performed. Approximately 8 h to 18 h passed until subsequent microbiological sampling of the rock samples in the cold room. Drilling fluids and seawater were collected at each site and stored at −80 °C for DNA analysis.

**Mineralogical and microbiological characterizations of thin sections of basalt cores**. To clarify mineral composition and microbial distribution within rock fractures, thin sections were prepared according to a protocol established for the localization of endosymbiotic cells in chemosynthetic animals[39]. Fracture-bearing core samples were dehydrated twice in 100% ethanol for 5 min, and core samples were infiltrated four times with LR white (London Resin Co. Ltd., Aldermaston, England) for 30 min and solidified in an oven at 50 °C for 48 h. Solidified blocks were trimmed into thin sections and polished with corundum powder and diamond paste. For the staining of microbial cells embedded in LR white, TE buffer with SYBR Green I (TaKaRa) was mounted on thin sections. After dark incubation for 5 min, thin sections were rinsed with deionized water, mounted with the antifade reagent VECTASHIELD (Vector Laboratories, Burlingame, CA, USA) and then observed using a confocal laser scanning microscope (IX71 with the FLUO-VIEW 300 system; Olympus) or an epifluorescence microscope (BX51; Olympus). Two ranges of fluorescence between 540 nm and 570 nm and 570 nm and 600 nm were used to discriminate microbial cells from mineral-specific fluorescence signals.

Mineral assemblages and textures were observed using an optical microscope (BX51; Olympus) and a charge-coupled device (CCD) camera (DP71; Olympus). Carbon-coated thin sections were characterized using a scanning electron microscope (S4500; Hitachi, Ibaraki, Japan) at an accelerating voltage of 15 kV. Back-scattered electron imaging coupled to energy-dispersive X-ray spectroscopy (EDS) was used to analyze the chemical compositions of mineral phases according to image contrasts.

To analyze microbial cells found in rock fractures by nanoscale secondary ion mass spectrometry (NanoSIMS) at Kochi Institute for Core Sample Research, JAMSTEC (NanoSIMS 50 L; CAMECA; AMETEK Co. Ltd., Gennevilliers, France), 3-μm-thick sections were fabricated using a FIB sample-preparation technique using a Hitachi FB-2100 instrument (Hitachi) with a micro-sampling system. The thin-section sample was locally coated with the deposition of W (100–500-nm thick) for protection and trimmed using a Ga ion beam at an accelerating voltage of 30 kV.

The elemental images of C, O, and N as CN, Si, P, S, and FeO from the FIB thin-section sample were obtained by an ion imaging using NanoSIMS 50 L ion microprobe. A focused primary $Cs^+$ ion beam of ~1.0 pA (100-nm beam diameter) was rastered over $12 \times 12$ μm$^2$ for U1365-12R2 and $16 \times 16$ μm$^2$ for U1365-8R4 on the samples. Secondary ions of $^{12}C$, $^{16}O$, $^{12}C^{14}N^-$, $^{28}Si^-$, $^{31}P^-$, $^{32}S^-$, and $^{54}Fe^{16}O^-$ were acquired simultaneously with multidetection using seven electron multipliers at a mass-resolving power of ~4500. Each run was initiated after stabilization of the secondary ion-beam intensity following presputtering of <~2 min with a relatively strong primary ion-beam current (~20 pA). Each imaging run was repeatedly scanned (20 times) over the same area, with individual images comprising $256 \times 256$ pixels. The dwell times were 10,000 μs/pixel for the measurements, and total acquisition time was ~3 h. The images were processed using the NASA JSC imaging software for NanoSIMS developed by the Interactive Data Language program[40].

After NanoSIMS 50 L analysis, 3-μm-thick sections were further thinned to 100-nm thick for examination using a JEOL JEM-ARM200F transmission electron microscope (JEOL, Tokyo, Japan) operated at an accelerating voltage of 200 kV at the Kochi Institute for Core Sample Research of JAMSTEC. X-ray elemental maps were obtained using EDS with a 100-mm$^2$ silicon drift detector and JEOL Analysis Station 3.8 software (JEOL) in scanning transmission electron microscopy (STEM) mode. A JEOL 2010 transmission electron microscope equipped with EDS at the University of Tokyo was also used to obtain EDS spectra from the 100-nm-thick sections.

**Contamination check**. To monitor contamination, fluorescence microspheres (0.5 μm in diameter) were used. This approach is not quantitative, but provides evidence of the occurrence of particle contamination, even in interior structures of basaltic samples (e.g., microfractures and veins). For the first step, a bag of fluorescence microspheres was placed on the core-catcher of each core, and after core retrieval and sampling, all microbiological samples were checked for the presence of microspheres.

Procedures involved in the evaluation and reduction of drilling-fluid contamination are schematically illustrated in Supplementary Fig. 1. Contamination was initially examined on the untreated exterior by removing small pieces of rock using a flame-sterilized hammer and chisel. The removed rock exterior was soaked in 25 mL 3% NaCl solution, and microspheres suspended in the NaCl solution were pooled in a 50-mL centrifuge tube. This procedure is necessary to confirm exposure of the cores to microspheres during drilling. The

rock surface was washed twice with 25 mL 3% NaCl solution in a sterile plastic bag. Small pieces of the washed exterior were removed with a flame-sterilized hammer and chisel, and wash solutions were pooled in a 50-mL centrifuge tube. After the washing step, the rock surface was flamed with a propane torch in a consistent manner (i.e., constant exposure time and distance between flame and core surface). The flamed rock was cracked open using a flame-sterilized hammer and chisel, and small pieces from the interior and exterior were separately soaked in 25 mL 3% NaCl solution, followed by pooling of the solutions in a 50-mL centrifuge tube. To enumerate microspheres, 3 mL of the aliquots were filtered using 25-mm black polycarbonate filters (0.22-μm pore size) and examined under epifluorescence using an Olympus BX51 microscope (Olympus, Tokyo, Japan). The minimum detection limit was determined to be ~100 microspheres/cm$^3$ rock based on the mean ± standard deviation (SD) of five replicate measurements of blank counting ($n = 5$).

**Community composition analysis**. Prokaryotic DNA was extracted from 0.1 g of the powdered inner core[41], which had been frozen at −80 °C for storage. Drilling fluids collected at each drilling site and stored at −80 °C were also subjected to DNA extraction. In 300 μL of alkaline solution (pH 13.5; 75 μL of 0.5 N NaOH and 75 μL of TE buffer), powdered core samples were incubated at 65 °C for 30 min, and the aliquots were then centrifuged at $5000 \times g$ for 30 s. After centrifugation, the supernatant was transferred to a new tube and neutralized by the addition of 150 μL 1 M Tris–HCl (pH 6.5). The DNA-bearing solution (pH 7.0–7.5) was concentrated using cold ethanol precipitation, and the DNA pellet was dissolved in 50 μL TE buffer and stored at −4 °C or −20 °C for longer storage. DNA extraction from subsamples was performed in parallel with one extraction negative control, to which no sample was added. The negative control was also subjected to pyrosequencing.

378-bp 16S rRNA gene region was amplified using the primers Uni530F and Uni907R for pyrosequencing using the GS FLX System sequencer[42] (Roche Applied Science, Penzberg, Germany). The primers were extended with adaptor sequences (Uni530F: CCATCTCATCCCTGCGTGTCTCCGACTCAG; and Uni907: CCTATCCCCTGTGTGCCTTGGCAGTCTCAG). The forward primer Uni530 was barcoded with 8-mer oligonucleotides to obtain sequences from multiple samples in a single run[43]. Thermal cycling was performed with 30 cycles of denaturation at 95 °C for 30 s, annealing at 54 °C for 30 s, and extension at 74 °C for 30 s. A polymerase chain reaction (PCR) amplicon with the expected size was excised from 1.5% agarose gels after electrophoresis and purified using a MinElute gel extraction kit (Qiagen, Valencia, CA, USA). DNA concentrations of the purified PCR amplicons were measured using a Qubit fluorometer with the Quant-iT dsDNA HS assay kit (Invitrogen, Carlsbad, CA, USA). The concentration of total dsDNA in each sample was adjusted to 5 ng/μL. Emulsion PCR was performed using the GS FLX Titanium emPCR kit Lib-L (Roche Applied Science) to enrich DNA library beads for the GS FLX System sequencer. Amplified DNA fragments were sequenced according to manufacturer instructions (Roche Applied Science).

Raw reads were demultiplexed, trimmed, and filtered based on their 8-bp sample-specific tag sequences, and quality values and lengths were assigned using the "clsplitseq" command in the MOTHUR program[44]. Filtered sequences were denoised based on sequence clustering, and possibly chimeric sequences were detected and eliminated using the MOTHUR program. The screened reads were aligned using the Greengenes reference dataset or the SILVA 128 database and the Needleman-Wunsch algorithm in the MOTHUR program. To assign sequences into phylotypes as operational taxonomic units (OTUs), the neighbor-clustering algorithm was employed using the MOTHUR program with a cut-off of 97% sequence similarity. Using the "vegan" package in R[45], nonmetric multidimensional scaling plots were obtained for phylotypes clustered with 97% similarity. Additionally, clustering and heatmap analysis of microbial communities were analyzed using the "vegan" package in R[45]. The phylogenetic affiliations of the phylotypes were analyzed along with closely related sequences retrieved from GenBank (http://www.ncbi.nlm.nih.gov/genbank/) through BLASTn searches (somewhat similar sequences) using the neighbor-joining method in the ARB software package[46]. The rapid bootstrapping algorithm (500 bootstrap replicates) in the ARB software package was then used to score the branching patterns in neighbor-joining trees.

**μ-Raman spectroscopy**. Raman spectra from fracture-filling minerals were obtained using a 50-cm single polychromator imaging spectrograph (Bruker Optics, Osaka, Japan), which was equipped with an optical microscope (BX51; Olympus), an $Ar^+$ laser (514.5 nm, 5500 A; International Light Technologies, Peabody, MA, USA), and a CCD camera (1024 × 256 pixels; DU401A-BR-DD; Andor Technology, Belfast, Ireland). The incident laser was operated at 20 mW, and the spatial resolution was ~1 μm. An edge-cut filter was used to remove the Rayleigh line. Raman lines of naphthalene at 513.6 cm$^{-1}$, 763.5 cm$^{-1}$, 1021.3 cm$^{-1}$, 1147.3 cm$^{-1}$, 1382.3 cm$^{-1}$, 1464.3 cm$^{-1}$, and 1576.3 cm$^{-1}$ were used to calibrate Raman shift. Analytical uncertainty typically ranged at ~1 cm$^{-1}$. To analyze iron-bearing phases, a non-dispersive filter was inserted to reduce the intensity of the laser beam by ~10% in order to avoid beam-induced transformation from goethite to hematite.

**Organic carbon characterizations of clay fractions of basalt cores**. The portions of rock cores examined as described above were powdered using a mortar and pestle. Clay-sized fractions in powdered samples were suspended in distilled and deionised water, centrifuged at 3000 rpm for 5 min, and then the supernatant was collected. Next, the clay fraction was collected from the supernatant by centrifugation at 10,000 rpm for 10 min and dried at 50 °C and then subjected to organic carbon characterizations. KBr pellets made of the clay fractions were analyzed with Fourier transformed infrared-ray (FT-IR) spectrometry (Perkin Elmer Spectrum 2000, Tokyo, Japan) to clarify lipids in extant microorganisms. The pellets in the specimen chamber filled by N₂ gas were analyzed by infrared rays through KBr beam-splitter with the MCT detector. Representative FT-IR spectra were obtained by averaging 100 individual spectra. The contents of organic carbon in the clay fractions were measured using a mass spectrometer (Thermo Electron DELTAplus Advantage; Thermo Fisher Scientific Inc., Waltham, MA) connected to an elemental analyzer (EA1112, Thermo Electron DELTAplus Advantage) through a Conflo III interface. The clay fractions were heated at 100 °C in 3% HCl to eliminate carbonate minerals, washed twice with distilled, deionized water, and dried.

**Reporting summary**. Further information on research design is available in the Nature Research Reporting Summary linked to this article.

## Data availability

Sequence data that support the findings of this study have been deposited in the DNA Data Bank of Japan (DDBJ) with the accession codes LC524012 to LC524122. All other datasets generated during the current study are available from the corresponding author on reasonable request.

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

## Acknowledgements

The authors thank the crew members, drilling team, technical staff, and shipboard scientists of drilling vessel *JOIDES Resolution* for their support during Integrated Ocean

Drilling Program (IODP) Expedition 329. The authors are grateful to K.J. Edwards for useful discussion. We also thank G.-L. Zhang and C. Smith-Duque for basaltic core descriptions during Expedition 329, and the technical staffs of the Kochi Institute for Core Sample Research, JAMSTEC. This is a contribution to the Deep Carbon Observatory (DCO) and the Earth 4D: Subsurface Science and Exploration, CIFAR. It is Center for Dark Energy Biosphere Investigations (C-DEBI) publication 526. This study was supported, in part, by the Japan Society for the Promotion of Science (JSPS) Strategic Fund for Strengthening Leading-Edge Research and Development (to JAMSTEC and F.I.), the JSPS Funding Program for Next Generation World-Leading Researchers (GR102 to F.I.). This work was also supported, in part, by the Astrobiology Center Program of National Institutes of Natural Sciences (NINS) (GRAB311023 to Y.S.) and JSPS KAKENHI Grant Numbers 18H04468, 18K18795 (to M. I.)

## Author contributions

Y.S. designed the study. Y.S. and S.M. collected and analyzed the basaltic core samples as shipboard scientists during IODP Expedition 329. Y.S., N.T., S.Y., H.M., and H.K. performed mineralogical characterizations, including microscopic observations. Y.S., Y.M., and M.I. conducted NanoSIMS analysis. Y.S., M.K., Y.A., and T.H. performed DNA sequencing and data analysis. S.D. and F.I. led the expedition. Y.S., M.K., S.Y. S.D., and F.I. co-wrote the manuscript. All authors discussed the results and commented on the manuscript.

## Competing interests

The authors declare no competing interests.
