## [Peer Review File · Communications Biology]

Reviewers' comments:

Reviewer #1 (Remarks to the Author):

This work is a significant contribution to the field of ocean crust microbiology. It advances our understanding of microbial habitats, potential food sources, and niches available for life in older crust. By drawing comparisons with microbes from another cool, oxic aquifer and giving updated estimates of microbial concentrations in ocean crust, the authors have significantly advanced the field. The manuscript is well written and they give multiple lines of evidence to back up their claims. Nice work.

Reviewer #2 (Remarks to the Author):

- Key results:

The relatively high concentration of primarily heterotrophic bacterial biomass associated with nontronite clay lining fractures in old, cold subseafloor basalt is likely due to adsorbed organic matter in the clay, which is of a magnitude higher than the surrounding mineral.

- Originality and significance:

This work is original, very well written and very interesting, with relevance to understanding the role of deep subsurface biology in the earth system. Conclusions are well supported by multiple lines of data and methods, with commendably thorough care taken over controls and contamination.

The following comments, I recommend considering:

Figure 4: In the interest of clarity, the caption would benefit from some additional explanation of what the schema is showing e.g. do the different arrow colours represent different flow kinds? Which panel shows substrate from seawater and which from rock?

L282: A modification to the statement of amino acid formation may be necessary to acknowledge that the difference in rock type and geochemical conditions between Lost City serpentine and old, cold basalt makes directly comparing them not particularly strong.

Reviewer #3 (Remarks to the Author):

The work by Suzuki et al reported their detection of microbial cells in the deep basalt crust by using multiple methods. Microbial cells have been reported presence in the relatively young basalt (mostly younger than 8Ma), but if microbes could manage to make a living in the older crustal environments remains an open question. This work presented compelling

evidences to show that microbes could survive in the much older (104Ma) basaltic crust, and they showed that the microbes in the older crust are dominant by heterotrophs who are fueled by organic carbon. This finding would have big influences when considering the scale of deep biosphere within crust, it may also trigger further research interestes to check if it's generally a habitat for microbes in other old cold crusts on Earth? and how similar or different are they? Where do they originate and how fast they evolve?.....

The paper is well written, easy to understand, the data and images are convincing. I would be very happy to see its publication in the near future. I only have a minor question for the authors:

The authors compare the microbes in the fluids of North Pond basement with those of their rock samples, as far as I know there're already reports on the microbial composition in the rock samples from North Pond, it may be more proper to make those comparison.

Reviewers' comments are black, whereas our responses are blue.

Reviewers' comments:

Reviewer #1 (Remarks to the Author):

This work is a significant contribution to the field of ocean crust microbiology. It advances our understanding of microbial habitats, potential food sources, and niches available for life in older crust. By drawing comparisons with microbes from another cool, oxic aquifer and giving updated estimates of microbial concentrations in ocean crust, the authors have significantly advanced the field. The manuscript is well written and they give multiple lines of evidence to back up their claims. Nice work.

We are very grateful to this reviewer for evaluating our work as a significant contribution to the field of ocean crust microbiology.

Reviewer #2 (Remarks to the Author):

- Key results:

The relatively high concentration of primarily heterotrophic bacterial biomass associated with nontronite clay lining fractures in old, cold subseafloor basalt is likely due to adsorbed organic matter in the clay, which is of a magnitude higher than the surrounding mineral.

- Originality and significance:

This work is original, very well written and very interesting, with relevance to understanding the role of deep subsurface biology in the earth system. Conclusions are well supported by multiple lines of data and methods, with commendably thorough care taken over controls and contamination.

We are also grateful to this reviewer for considering that conclusions are well supported by multiple lines of data and methods.

The following comments, I recommend considering:

Figure 4: In the interest of clarity, the caption would benefit from some additional explanation of what the schema is showing e.g. do the different arrow colours represent different flow kinds? Which panel shows substrate from seawater and which from rock?

Thank you very much for pointing out some explanations for arrows and their colors. We added the explanations for clarity.

L282: A modification to the statement of amino acid formation may be necessary to acknowledge that the difference in rock type and geochemical conditions between Lost City serpentine and old, cold basalt makes directly comparing them not particularly strong.

As the ~1.4 km rock sequence recovered at Lost City Site (U1309) was dominantly gabbroic, only a few percent of the drill core consisted of ultramafic rock. Gabbro and basalt are both mafic rocks and geochemically similar in composition. Therefore, there might not be so much geochemical difference between our samples and the Lost City sample. For revision, we simply described that the Lost City sample was obtained from gabbroic basement.

Reviewer #3 (Remarks to the Author):

The work by Suzuki et al reported their detection of microbial cells in the deep basalt crust by using multiple methods. Microbial cells have been reported presence in the relatively young basalt (mostly younger than 8Ma), but if microbes could manage to make a living in the older crustal environments remains an open question. This work presented compelling evidences to show that microbes could survive in the much older (104Ma) basaltic crust, and they showed that the microbes in the older crust are dominant by heterotrophs who are fueled by organic carbon. This finding would have big influences when considering the scale of deep biosphere within crust, it may also trigger further research interestes to check if it's generally a habitat for microbes in other old cold crusts on Earth? and how similar or different are they? Where do they originate and how fast they evolve?.....

The paper is well written, easy to understand, the data and images are convincing. I would be very happy to see its publication in the near future. I only have a minor question for the authors:

The authors compare the microbes in the fluids of North Pond basement with those of their rock samples, as far as I know there're already reports on the microbial composition in the rock samples from North Pond, it may be more proper to make those comparison.

We added data from Zhang et al. (2016) that are showing the dominant colonization of *Alteromonadales* members in rock core samples. The dominance of *Alteromonadales* members is consistent with our rock core sample and the North Pond fluid sample. However, the dominance of *Campylobacterales* members was not consistent between the North Pond rock core and fluid samples and between the North Pond rock core and our rock core sample. As it is very difficult to discuss causes of this discrepancy, we simply state that the nature of the basement fluid may be very similar in relatively young (8-Ma and 13.5-Ma) basaltic basement in both the Atlantic and Pacific Oceans.

REVIEWERS' COMMENTS:

Reviewer #2 (Remarks to the Author):

Reviewer comments have been adequately addressed. Recommend for publication.

Reviewer #3 (Remarks to the Author):

I have no other comments, nice work.